# What does it mean to be the main caregiver to a terminally ill family member in Lithuania?: A qualitative study

Jolanta Kuznecovienė[1]*, Rūta Butkevičienė[1], W. David Harrison[1], Eimantas Peičius[1], Gvidas Urbonas[1], Kristina Astromskė[2]

**1** Department of Bioethics, Lithuanian University of Health Sciences, Kaunas, Lithuania, **2** Department of Health Management, Lithuanian University of Health Sciences, Kaunas, Lithuania

* jolanta.kuznecoviene@lsmuni.lt

## Abstract

### Introduction

Family caregivers are a great resource for providing dignified end-of-life care for terminally ill patients. Framed from the perspective of role theory and the relational nature of providing and receiving care, study objectives were as follows: (1) to capture caregivers' understanding of the process of taking on the role of main caregiver, (2) to conceptualize their understanding of the functions that they assume while being the main caregivers, and (3) to understand how they experienced the consequences they confronted.

### Methods

The research team employed the methodological strategy of descriptive thematic analysis using a semi-structured interview guide. The sample consisting of 33 family caregivers was recruited using purposeful and snowball sampling strategies in 2020. Interview data was analyzed using content-driven inductive thematic analysis.

### Results

The data analysis revealed four main themes that structure the process of becoming the main care giver of a terminally ill family member and the meaning of the caregiver role: (1) inaccessibility and mistrust of public care services for persons with terminal illness, (2) moral obligations and responsibilities of immediate family and friends, (3) cultural traditions, (4) the caregiver feels responsible for everything. The themes describe the social role of family caregiver in social context, address the process of taking on the role of caregiver and living with systemic corruption.

### Conclusions

Recognition of caregiving experiences is essential in planning better systems, in direct practice and in confronting corruption. The study suggests the need for open communication, accessibility of quality services, and the recognition of caregivers as care-team members.

**Data Availability Statement:** All relevant data are within the manuscript in the result section and in S2 File.

**Funding:** This study was supoported by Lithuanian Science Foundation, Grant Agreement no. S-GEV-20-2. The funders had no role in study design, data collection and analysis, decision to publish, or preparation of the manuscript.

**Competing interests:** The authors have declared that no competing interests exist.

The larger implication is that the increasing numbers of distressed caregivers and aging populations can be considered as public health populations, and thus addressable through public health methods.

## Introduction

There exists today a worldwide examination of what it means to live and to die with dignity. Philosophical, ethical, psychological, and cultural considerations about death and dying have become common topics in caring professions and in many quarters of the public. The study is one component of a larger project concerning the meaning and promotion of dignity at the end-of-life [EoL], embracing professional and lay perspectives. Dignity was conceptually defined as an individual's intrinsic sense of worth coupled with respect on the part of the self and others. This component study was based on the assumption that there is a relationship between the caregiver's experience and the dying individual's sense of dignity. Caregivers were the focus of the study not only because of their relational importance in the establishment and reinforcement of a dying person's sense of dignity, but also because the experience of caregiving can promote or threaten the caregiver's own sense of dignity.

Studies show that most patients prefer spending the last period of their lives at home [1]. Helping patients to die at home, if possible, is seen as part of "humane healthcare" [2]. However, increasing proportions of hospital deaths [3, 4] show that patients' expectation to choose a comfortable setting for their end-of-life (EoL) is often not met. Respecting patients' wish to spend their last days at home can be seen as preserving patients' dignity and assuring "good death" among their families [4, 5]. Official Lithuanian health policies concerning quality and continuity of care are fairly consistent with others in the European Union. However, many policies are most accurately seen as aspirational rather than implemented. Lithuania represents a country in which much of the direct, hands-on care and medical decision-making are components of informal caretaker roles, in contrast to places where care teams involving specialists working in partnership with informal caregivers are more common. Moreover, Lithuania exemplifies the problem of corrupt negotiation of roles by informal caregivers.

Family caregivers are a great resource for providing dignified end-of-life care to terminally ill patients. While a full review of the large body of caregiver research is beyond the scope of the current study, multiple empirical studies and meta-analyses have already focused on the ways the family members get involved in the caregiving process [5–7], on the roles and functions they take [8, 9], on their needs for support [9–11], and on outcomes [12] they experience while caring for their relatives at the end of life. Of particular note are the contributions of Kristjanson and the idea that families constitute "hidden patients" [8]. In a study of the needs for support of caregivers, Hudson, Aranda, and Kristjanson identified barriers that must be confronted in order to support caregivers, emphasizing the need for communication [13]. However, despite many studies aimed at improving caregivers' experience and effectiveness of their care, the extraordinary importance of informal or family caregivers in healthcare systems at stages similar to Lithuania's is sometimes underestimated. For example, in a major literature review of empirically-tested models of palliative care in Europe, Siouta and colleagues documented the benefits of practice based on varied team approaches in different countries [14]. In all the studies reviewed, professionals constituted the teams for integrated palliative care. Non-professional caregivers were not reported as members of these teams. They were mentioned only as embodiment of dependent variables, subjects of the desirable outcomes, namely,

improved communication and reduced caregiver burden. Their role as team-members or caregivers was essentially absent. Similarly, Veloso and Tripodoro concluded that the role of family caregivers is too often ignored in considerations of health care policy, and it is often seen as a private, family matter that obscures both its importance to the quality of care and the burdens that the caregiver experiences [15]. Meanwhile, in order to promote collaboration in the care of the patient and provide bereavement support, the needs of the family and other close caregivers throughout the course of the illness must be recognized and attended to [11]. It was recognized that more active involvement of family is highly important in EoL decision-making [16, 17] and care [2].

How does one take on the caregiver role for someone who is dying? What does it mean to become an end-of-life caregiver in Lithuania? Lithuania is experiencing an ongoing shift toward contemporary Western European cultural and professional practices. The 1992 Constitution includes aspirational statements about national values and where the country aims to be in regard to care. Article 38 states that "The duty of children is to respect their parents, to take care of them in their old age, and to preserve their heritage." Article 53 states that "The State shall take care of people's health and shall guarantee medical aid and services for the human being in the event of sickness . . . free of charge at State medical establishments" [18]. These obligations leave a gap between formal and informal care. On the one hand people can get formal services of palliative care in state nursing hospitals, but the level of services, as studies show, are not always in line with patients' expectations [19]. On the other hand, Lithuania's National Strategy for overcoming the consequences of aging population seeks to promote the involvement of the informal sector (family, relatives, neighbors) in the long-term care for elderly [20]. However, studies revealed that relatives are not able to care for the sick alone and need professional help to ensure both the basic needs of the patient and the support of loved ones caring for their family members. Integrative care services are rather new, they are not available to everyone, and not everyone is familiar with them [21]. Family members who care for severely ill patients are at risk of exhaustion and deterioration of their health over time. They need not only counseling on nursing issues, but also emotional support and help in combining family and work responsibilities [22]. Informal care giving is a common phenomenon in Central and Eastern Europe (CEE). The lack of support for informal caregivers in particular, and the need for integration of EoL care seem to be urgent in CEE palliative care context [23]. For example, Baji and colleagues demonstrated the impact of demographic changes in Poland and Eastern Europe, and changes to the familial system of care for the elderly and dying urge the necessity for action in EoL care [24]. The unmet needs in all constructs decreased quality of life of family members and increased the care burden [25].

The qualitative study reported here is set in the context described above. It addresses the experiences and perceptions of informal care providers in a health care system with palliative care services that are in a comparatively early stage of development. These are the people who are confronted with complex roles including both the physical aspects of decline and death and questions about how to best provide this care in a way that enhances the dignity of the individual who is facing the end of life. The study is framed from the perspective of role theory and the relational nature of providing and receiving care. The ways that the roles are learned, invented, enacted, and experienced constitute the core of the current investigation. For most care providers, the process of taking on the role and providing for physical and emotional care is tacit. Through the study of narratives of caregivers in a semi-structured format, many of the important tacit elements are made explicit in order to learn how to enhance the experience for both caregivers and those who are cared for.

Much of the existing body of research on families, caregivers, and palliative care has been grounded in concepts and process conceptualized from role theory. Building on conceptual

frameworks embodied in these works, the current study is organized around ideas from role theory, which has a substantial sociological grounding going back to Merton's original formulation [26], but also an intuitive grounding in everyday life that makes it useful in qualitative discourse. A multidisciplinary team (sociology, social work, law, bioethics, public health) of researchers designed the current study which aimed to capture caregivers' understanding of the process of taking on the role of main caregiver in a healthcare system with developing palliative care services, to conceptualize their understanding of the functions that they assume while being the main caregivers, and to understand how they experienced the consequences they confront as the process evolves and finishes. The team's aim was to provide insights that would be relevant not only to Lithuania, but also to the Central Eastern European Region as well due to similar context and issues.

The three objectives of this qualitative study were the following: (1) to capture caregivers' understanding of the process of taking on the role of main caregiver, (2) to conceptualize caregivers' understanding of the functions that they assume while being the main caregivers, and (3) to understand how they experienced the consequences they confronted in enacting the role.

## Materials and methods

### Research strategy

The research team used the strategy of applied thematic analysis as presented by Guest, et al. [27] to structure the study. Applied thematic analysis is flexible and does not propose a methodological recipe. The strategy is integrative in that it allows for the pursuit of phenomenological understanding through the use of data collection and analysis methods derived in part from a positivistic tradition. Interview text is data observed and recorded in the real world, meaning that it is positivistic data. The analysis process was designed to seek commonalities of experience reflected in this data, and to apply an interpretive, phenomenological orientation to the systematic assessment and understanding of the essential meaning conveyed by the data. The team used the interpretive phenomenological analysis method advocated by Pietkiewicz and Smith [28]. Although Pietkiewicz and Smith did not identify the use of positivistic data for interpretive elements explicitly, they describe the use of interview (and other empirical, positivistic) data as foundational elements to construct a phenomenological presentation of how people understand a situation. The logic and procedures of the study are shown in Table 1.

### Interview guide

Qualitative methods of this type are particularly useful in capturing content and context that may not have been known or hypothesized a priori [27]. To find themes the team of six public health faculty members designed an interview guide that provided preliminary structure for data collection. The guide built upon the content of previous studies of caregivers' behavior, many of which were framed explicitly or tacitly by the concept of social role. For example, Stajduhar et al. reviewed dozens of studies that emphasize the knowledge and skills of the caregiving role and how caregivers acquire them [11]. The intent of the guide was to explore the experiences of caregivers in fulfilling the caregiver role, including role elements concerning everyday life of caregiving, the needs of the terminally ill person and the family, patterns of communication and decision-making with professionals, and caregivers' perceptions of a dignified end-of-life experience. Team members were sensitive to the changes that one might experience in the role over time. Interviewers used probes to ensure that the data were characterized by clarity, sensitivity to difficult topics, openness to important new topics, and explanations of inconsistencies [29].

**Table 1. Logic flow of the study.**

| Phase | Research activity | Participants |
|---|---|---|
| Phase 1 | Project questions are clarified, literature is reviewed and shared. | JK, RB, WDH, GU, EP, KA |
| Phase 2 | Sample, interview guide, data analysis, form of findings are planned. | JK, RB |
| Phase 3 | Data collection, analysis, and verification. | |
| | 1. Three interviews are conducted, transcribed, coded independently by each of two interviewers. Interviewers compare codes and data, compiling initial shared coding after three interviews. Subsequent interviews conducted, transcribed, coded with initial and new codes (N = 27). Codes are translated from Lithuanian to English. | RB, JK |
| | 2. Interviewers share codes, construct categories based on shared meaning of codes. Categories are compared and refined into themes through memos and discussion by interviewers and two additional research team members. | RB, JK, WDH, GU |
| | 3. Independent review of data and themes for consistency and coherence done by two additional team members. | EP, KA |
| Phase 4 | Discussion and organization of findings and conclusion. | WDH, JK, RB, GU, KA, EP |
| Phase 5 | Preparation for publication. | WDH, GU, RB, JK, KA, EP |

Note: JK—Jolanta Kuznecovienė, RB—Rūta Butkevičienė, WDH—W David Harrison, EP—Eimantas Peičius, GU—Gvidas Urbonas, KA—Kristina Astromskė

The original interview guide is available as S1 File. Discussion to refine the early drafts of the guide and informal pre-tests led to the decision to reduce the number of stimulus questions and to frame them to allow for a broader range of responses. After analysis of the first interviews, the researchers supplemented the guide with emerging questions on topics about knowledge and use of social services and other formal and informal social supports, the aversion to and fear of using any sort of inpatient palliative care facilities, and the necessity to provide unofficial payments to medical personnel in order to receive any services. Consistent with the design of the study, these topics reflect specific Lithuanian realities that may be clearer in the country's specific national context than they would be in different settings where they are less apparent to practitioners, but still present. The questions stimulating by far the most valuable data were "How do you personally understand dignity at the end-of-life?" and "What factors ensure dignity at the end of life?".

## Sample size and characteristics

In his classic text on interviewing, Spradley emphasized that the selection of informants should be guided by the ability and willingness for informants to share important, complex, and emotional information [30]. In research intended to understand sensitive, sometimes intimate topics, this task typically requires purposeful selection and a high degree of ability to engage and communicate on the part of researchers. The researchers in the current study used these ideas as a guide in constructing an appropriate sample through integrated purposive and snowball methods of engagement. The purposeful and snowball strategies are appropriate in health research "for the identification and selection of information-rich cases related to the phenomenon of interest" [31]. The snowball component was used to identify individuals by "sampling people who know people that generally have similar characteristics who, in turn know people, also with similar characteristics" [31].

Two criteria were used in recruiting and screening the sample members. The criterion of being an adult who had cared for a loved one who died of an incurable disease was important in order to learn about a full cycle of assuming the caregiver role, actively providing care, and ending in the caregiver role. The second criterion was that the loved one's death had occurred within the previous five years. This was important to take into account the changing context of services and social conventions. Potential informants were invited to participate in the research by responding to an announcement on the websites of organizations that brought together relatives and persons with incurable illnesses in 2020. Informants were also engaged through personal contacts and by asking interviewees to name a familiar person who had experience in caring for a loved one suffering from an incurable disease at the end of his or her life. The interviewers found their superficial recognition or familiar relationship with 16 participants to be particularly valuable in engaging them and acquiring snowball referrals. All potential participants were contacted by telephone, screened, and invited to participate if they met the study criteria. This selection process produced no drop-outs from the study. Twenty-eight in-person interviews were conducted privately at a location chosen by the informants, at their home or workplace. Six interviews were conducted via audio visual software at locations chosen by the participants. Interviews ranged from 45 min to 88 minutes. Recruitment continued until the researchers judged that saturation had been reached, and significant repetition of codes and categories was occurring. The sample allowed the researchers to emphasize the fundamental commonality of informant experience while allowing for variations in order to fully develop themes from the data.

The researchers concluded that substantial saturation of content categories was reached after 33 interviews, at which point interviewing stopped. This was determined by the research team's high degree of agreement that data that were being consistently coded in specific groups or categories, and that new coding was not repeated significantly.

Table 2 profiles the participants, 29 of whom were women. Six were over 60 years of age, and the majority were middle-aged. Most of the caregivers (N = 20) were adult children of the person being cared for. The other caregiver relationships to the ill person were as follows: spouse (N = 9), sibling (N = 2), adult grandchild (N = 2). Twenty-two of the 33 people the sample group cared for had a primary diagnosis of cancer. Twenty-three were sixty years of age or more. In 25 out of 33 cases, caregiving took place primarily at home.

Prior to the study, the research project was approved by Kaunas Regional Ethics Committee of Biomedical Research in Lithuania [No. BE-2–86, issued 23–07-2020]. The aim of the study

**Table 2. Profile of caregivers participating in the study.**

| Characteristic | | N |
|---|---|---|
| **Gender** | Female | 29 |
| | Male | 4 |
| **Location of care (death)** | Home | 25 |
| | Hospital | 8 |
| **Age** | $\leq 40$ | 2 |
| | 41–60 | 24 |
| | $\geq 61$ | 7 |
| **Relation to person cared for** | Spouse | 7 |
| | Child | 20 |
| | Grandchild | 2 |
| **Interview media** | In person | 28 |
| | Video | 5 |

was explained personally to each participant including the meaning of terms, possible inconveniences involved in participation, and a guarantee of confidentiality and data protection. Each informant's written consent was obtained, or in the case of the five interviews conducted using audio-visual software because of pandemic conditions, video recordings were made and archived. The practice of video recording of consent to participate in research is relatively new. The researchers and ethics experts concluded that the interviewers could make use of the media, consistent with earlier successful large-scale applications of the practice [32]. The use of this method of obtaining informed consent proceeded smoothly, and without apparent issues.

It is noteworthy that five research interviews were conducted through audio-visual media because of the COVID-19 pandemic and public health orders to avoid in-person contact. The interviewers proceeded cautiously until they were convinced that the media were not adversely affecting data collection. In fact it was apparent in some cases that participants were already accustomed to sharing personal information in this way, or at least they were essentially uninhibited. In discussing the experience with participants, some expressed that they may have spoken more freely or discussed some topics more quickly because of the medium of the interview. The effects, if any, of media in collecting sensitive data remains a matter for further research. Sample of raw interview text are shown in S2 File.

## Data analysis

Interview data was analyzed using content-driven inductive thematic analysis. Essential categories were detected by recognizing and coding discrete passages of communication and then building them into categories and subsequently building categories into themes that were based on shared or contrasting passages. This type of analysis does not depend on predetermined codes or categories, but instead extracts them from data that is generated by the researchers and informants, who are most often engaged using purposive sampling [27].

In the current study the analysis proceeded in the three stages, outlined in Table 1, Phase 2. First, data was recorded and transcribed by the interviewers shortly after each interview was conducted. After the first three interviews, the interviewers developed a common list of codes from their data. Subsequently, the remaining transcripts were shared between the two researchers and coded using the prepared code list. At this stage, additional codes were added to the list. The second stage was the process of organizing codes into categories. These categories often involved constructs composed of collective codes and recorded meanings. The field notes made by interviewers were also used in the process of data analysis and interpretation. At stage three, the categories were compared, blended, and refined into themes. The process is illustrated with the development of Theme 1 in S1 File, entitled Progression from sample of interview text to codes, categories, and theme.

## Quality assurance

Four members of the research team discussed the data and its analysis at the beginning of this stage (Phase 3, Stage 2 from Table 1). Different insights and disagreements in understanding and interpretation were reconciled and the results were agreed upon. In order to strengthen trustworthiness of the findings, the other two team members independently reviewed the data and evolved themes at stage three. While the names of some codes, categories, and themes varied, there was a high degree of agreement on substance. The consistency and compatibility of findings between the data analysts made reconciliation possible with a high degree of concordance. The themes were also presented for discussion by medical professionals during a round-table seminar organized by the research team. Despite a number of digressions based on specific findings, the round-table group concurred that the findings were consistent with

**Table 3. Themes and categories.**

| Themes | Categories |
|---|---|
| 1. Inaccessibility and mistrust of public care services | 1a. Lack of awareness of public care services |
|  | 1b. Unavailability of public health care services |
|  | 1c. Patients, family and friends view services negatively |
| 2. Moral obligations and responsibilities of immediate family and friends | 2a. Doing the right thing |
|  | 2b. Shortage of immediate family and friends |
|  | 2c. The caregiver's professional qualification and previous experience |
|  | 2d. Emotional relations between care giver and ill family member |
|  | 2e. Friends and relatives lack care and communication skills |
| 3. "It's our generation": cultural aspects |  |
| 4. The caregiver feels responsible for everything | 4a. Practical nursing issues |
|  | 4b. Consulting and informing |
|  | 4c. Managing |
|  | 4d. Being together: providing emotional support |
|  | 4e. You have to stay there: negative experience of hospitalization |
|  | 4f. Exhausted physically and emotionally |
|  | 4g. Confronting consequences |

the realities that they faced or had experienced, even though they were understood from quite different viewpoints. Table 1 and S3 File constitute an audit trail for the data analysis process.

## Results

Four themes and related categories were found to structure the process, role, and meaning of becoming a main caregiver. These are presented in Table 3. The main themes are as follows: (1) inaccessibility and mistrust of public care services for persons with terminal illness, (2) moral obligations and responsibilities of immediate family and friends, (3) cultural traditions, (4) the caregiver feels responsible for everything.

### 1. Inaccessibility and mistrust of public care services

Palliative care services in Lithuania are not as well developed as in some other countries. The data shows that family caregivers use services in very limited ways. The categories that grounded this theme were consistently and prominently shared by informants.

**1a. Lack of awareness of public care services.** Many informants were unaware of services that might be useful. Participants stated that no one informed them about this possibility. Medical personnel do not provide information on what kind of state support they can receive in caring for a relative, their right to service, and the legal acts that regulate services. One informant described the situation concisely.

> *End-of-life care is clearly the most difficult period, because you are completely alone, because there was no help from official medicine, neither psychological nor social, absolutely none. . . . It wasn't like they would say, for example, you'll have a hard time. . . . If you need medication . . . painkillers, contact this or this specialist. If you are working, then maybe you need someone to be with the patient, then contact there, they will help you find care. Whether or . . . let's*

*say, maybe you'd like a psychologist or a clergyman to talk to the patient. . . . There were no such questions and suggestions at all. (Wife 55)*

As if continuing this story, another informant shared that she did not manage to receive the state financial support for caring for a loved one at home that she was entitled to because she received this information just a few days before the loved one died. Informants shared various examples of their care experiences related to the lack of information about public palliative care services and reflected on how the lack of information indirectly diminished the quality of care of their terminally ill relative:

*In a word, there is no respect for a person—neither for that patient nor for the caregiver. There are some laws somewhere, but they are somewhere in sky. To the particular person who needs them, they do not appear. They have to look for themselves, and in order to look, they still have to know where to look, that is. That's what I think, as soon as such a diagnosis is made for somebody, he needs to get comprehensive information on where and what questions you should turn to. It is then possible to plan care time in the best way possible for both the patient and the nurse. (Wife 57)*

Another informant, talking about lack of information about public care services, recalled that while caring for a loved one, she was unaware of the existence of palliative nursing teams providing professional nursing services, and was unaware that the provision of these services was reimbursed or partially reimbursed by the state. One informant accidentally learned from a conversation with an acquaintance that palliative care teams exist:

*I accidentally talked to a former student about palliative care and she says, look, I have established such an enterprise for palliative care and . . . then, we can arrange the paperwork and it is free, it is state reimbursed and we can try to help you. And so, these people who came through personal contacts, were of gold value. (Wife 51)*

**1b. Unavailability of public health care services.** Informants talked about inaccessibility of palliative care of all sorts, especially about long waiting lists for those who want to use service. The majority of participants discussed bribes given to the service administrators for shortening the waiting time to enter facilities. One participant described this as an "agreement."

*And nursing hospital, if without any such attempts to agree in some way with administration, then it puts you in line and you wait in line until someone from there dies. It's full absolutely [until then]. (Sister 65)*

**1c. Patients, family and friends view services negatively.** Another explanation for why one family member takes on almost all responsibility for caregiving is the refusal by the patient's family to use the services provided by long-term care facilities. Most informants reported that the possibility of caring for a sick close relative in an inpatient setting was deemed unacceptable by all family members, including the patient. Most of them explain that their refusal to make use of facility is based on strong distrust in the quality of services provided there:

*And in the institutions, it's just that the hypnotics are injected so that the patient doesn't ask for anything. (Daughter 45)*

## 2. Moral obligations and responsibilities of immediate family and friends

**2a. Doing the right thing.**   It was widely perceived that assuming the caregiver role was the right thing to do in certain circumstances. Caregiving was a duty that one should carry out in relation to one's relationship to the person who is ill. Several elements constituting this theme were found.

Some informants focused their caregiving responses and accounts on obligations and responsibilities. Informants reported incidents when a sick family member refused to go to the nursing hospital where patients commonly stay until they die. In most cases, the refusal to go to the nursing hospital was expressed very categorically, and the family simply carried out their will, saying during the interview that it was their obligation and responsibility to take care of the loved one at home.

In other cases, terminally ill loved ones shared their feelings of being scared by hospitalization, and caregivers just fulfilled their wishes:

*I lived with my grandmother until adolescence . . . She was very scared to die in the hospital, kept asking for home. My mother, her daughter, was also sick, so we advised that I take [grandmother] to me and, in exchange for my mother, take care of her. I can't imagine strangers doing all these intimate things to my grandma . . . I think that all people expect to die at home. (Granddaughter 49)*

**2b. Shortage of immediate family and friends.**   According to research data, a quarter of our informants assumed the role of the main caregiver as a consequence of their life circumstances. For example, they became caregiver when the caregiver was the only close family member who could take on the role, when other family members lived abroad, or when the ill person's relationships with other family members or relatives were severed.

**2c. The caregiver's professional qualification and previous experience.**   Becoming the main caregiver was made likely by an individual's education and profession, particularly if it related to medicine such as nurse or pharmacist. In at least four cases, this family member had taken responsibility for the health issues of the family members in the past, and in the current circumstances it seemed self-evident that this person should give care.

*My family handed over everything related to his care to my hands. But that surrender also means responsibility. They're about "you're in medicine, you know everything best, you're in order." . . . I'm used to it. It has long been my responsibility. I accepted it as normal . . . It goes without saying that it was me. (Daughter 47)*

**2d. Emotional relations between care giver and ill family member.**   Some informants revealed that one of the main motives that prompted them to take on this role was their emotional relationship with a sick loved one:

*We were so tightly tied up to each other that somehow it even seemed natural that only I could do it . . .. (Daughter 45)*

The prominence of filial responsibility and marital relationships between caregiver and the one cared for was coupled with the natural emotional attachments. Approaches to decisions about the best for the ill person to be taken care of varied among the caregivers too. For example, in some cases a family considered institutional care as an appropriate option for the terminally ill patient, but accepted the will of ill persons to be cared for at home as their duty and responsibility:

*Somehow at home we all talked it in one direction, to go to the hospital. But yeah, it was really very difficult with her. . . . When the disease was so advanced that it was no longer possible to stay at home, and she was not mentally inadequate and she did not sign up for a hospice, she did not agree on anything. (Daughter 57)*

**2e. Friends and relatives lack care and communication skills.** One of the informants' explanations for what determines who takes on the main caregiving role is that over the course of the illness, quasi-nursing care and communication with the ill person become more difficult. Analysis of the informants' narratives highlighted how the number of family members and close friends caring for a sick relative eventually is reduced to a minimum, frequently to one person.

*Because people come and knock even more out of balance. And even those glimpses of pity, words of pity, movements of pity like, say, show that everyone already knows that the person is already doomed. And then the person closes himself. Finally, she didn't want to communicate with anyone. Her will was for no one to go [visit]. (Husband 65)*

*. . . people are not taught to be with another person's suffering. It's so. (Daughter 55)*

Another informant, while describing behavior and relationships of other relatives and friends with a terminally ill person during his or her intensive care period, used the word "frustration". People wanted to do or say something useful or meaningful, but did not know how or were afraid of doing something wrong. Indeed, research data shows that frustration often becomes one of the most important elements in constructing a terminally ill person's relations with his or her relatives and friends. Many informants noted the frustration experienced by friends and relatives when they were with a sick and powerless loved one, the fear of the impending loss of a loved one, the fear of harm due to a lack of caring skills, the fear of not being able to communicate with a terminally ill patient in a proper way. Based on the research data, it could be assumed that these concerns are why some relatives reduce their visits, others stop attending the patient altogether, and others contribute to caregiving infrequently and unsystematically, for example by helping the main caregiver by shopping or driving the caregiver or sick relative to one or another institution. However, as the informants said, there may still be one person, embracing both the moral imperative and the ability to care and communicate:

*And the close people who used to interact with him have now backed down. And only then did I realize they didn't know what to do in that situation. They don't know, they're afraid of pain, they're afraid to touch pain, they don't know what to talk about, they just put it in the bushes, that's to say nowhere. For me, that suffering is my own and I cannot escape from it and I do not want to escape. It is good for me to do good to a loved one, to do as he needs to, as I understand, how we feel we need to do. This is my situation. And for others there is no need to worry, a burden they do not know how to lift, do not know what to do with it. (Wife 55)*

The inability to care and the emotional discomfort experienced by friends and relatives, and especially by the sick person, has significant consequences for the quality of life of both. Most of our informants said that their loved ones in the last months of intensive care limited the circle of people with whom they wanted to communicate to a minimum. In many cases only the dying person and the caregiver remained:

*But usually after a while it remains only one person who is caring, the main one. Because the whole family will not take care of loved one, because it is too difficult for them, and the habits need to be known. So one person remains. And that loved one also wants to be with that one person. He doesn't really want to see other people. My husband didn't even let other people take care of him. That's right, I still draw the following conclusion from my experience in the finale—there is only one person who cares. The main remains, everyone else comes, visits, leaves. (Wife 61)*

### 3. "It's our generation": Cultural aspects

Becoming a main caregiver is inseparable from cultural traditions. The caregivers participating in the study belong to a generation that cares about old traditions that say where and how terminally ill and dying family members should be cared for. This tradition means that a duty of a good child or spouse is to respond to the ill family member's will and expectations.

Informants noted that caring for the terminally ill relative at home is a feature of their generation, which meant the typical caregivers' older generation. Some informants do not expect such care from their own children. Even some terminally ill persons asked their children (caregivers) "to live your own life."

*My children won't take of me in the way I did . . . The idea of putting your parents to nursing hospital never would come to the mind of our generation. Unless your mutual relationship is broken. But for my children it wouldn't be any problem. (Daughter 59)*

*I was still working at that time and my brother helped me at home to take care of our mother. Mother kept saying "don't leave your job." I took three weeks of vacation, and at the end of vacation I said that I will try to prolong vacation by asking for unpaid days. She repeated several times: "Please, don't give up your job." (Daughter 52)*

Twenty-nine of the 33 participants in the study were women. The female roles of wife, daughter, mother traditionally include responsibility for family health and social care issues. Society and family "delegate" this responsibility, and women take it:

*My husband did not have that feeling of guilt/inconvenience that he needed to be taken care of. He would say you as a saint, but as if necessary. (Wife 59)*

### 4. The caregiver feels responsible for everything

This study revealed the complexity of expectations and responsibilities constituting the complex role of caregiver. The nature and intensity of these role components changed in the process of caring. This typically meant that role components were cumulative and new elements had to be enacted simultaneously with earlier ones. These role components had both practical and emotional aspects. Some caregivers described themselves as breathing straws for a sinking patient and for whole families.

**4a. Practical nursing issues.** Depending on the patient's condition, the process of caregiving required a range of nursing skills. Some informants had appropriate education themselves, or somebody in the family knew how to perform nursing actions as well as where and how to obtain needed nursing equipment or facilities. Those who did not have nursing knowledge

and skills tried to learn independently by observing nurses in clinical settings. They found where and how to get nursing resources and skills.

*I learnt myself to do intravenous infusion at home. Every day and night I did those infusions. . . . The only thing I didn't know how to do was how to insert a urinary catheter. (Wife 46)*

*When I took her home from hospital, I put her to my bed, but I saw the big pain she suffered. So, I found announcement in internet about functional bed, special mattress and rented all things needed. (Daughter 37)*

**4b. Consulting and informing.** The caregivers reported that they were responsible for communication between medical specialists, patient, and family. The study revealed that in some cases the caregiver was the the first and only person with whom physicians shared the patient's diagnosis or from whom they asked permission for the diagnosis to be shared with the patient. The physician sometimes discussed treatment options and plans with caregivers. The doctors sometimes asked caregivers to make decisions concerning treatment. During the entire journey of illness, the caregivers found themselves consulting with physicians.

*Hematologist first invited me and gave his diagnosis. And asked if she could speak about that with my husband. (Wife 46)*

*Oncologist asked me what treatment we chose: traditional or alternative. I knew what traditional means; they explained but I wasn't sure about alternative . . . so I was looking for contact, finding people who maybe knew more. (Wife 46)*

*I used to call oncologists and ask should I offer food to her when she refuses . . .. Sometimes I got very angry because Mom refuses to take medicine, I didn't know what to do. I asked her physician how should I speak to Mom to convince her to take it. . . . And what to do in case of pain. What medicine would help? You know, everything is new, you don't know what to do. (Daughter 52)*

Furthermore, the caregiver became responsible for giving information to the patient and other family members. In an effort to protect the patient and family from bad news, sometimes caregivers buffered the content of the message. For example, one informant told the patient and family that the cancer was classed stage one instead of stage three. Several reported hiding information since they did not know what to do with it.

*We didn't know if he [husband] understood his health condition. I didn't want to ask if he knew what is going on. What if he asks "what is going on with me? What to say? Keep lying? (Wife 68)*

**4c. Managing.** Even in the case of involvement of other family members or family friends, caregivers had to manage the functions and responsibilities of the complete support network—who would do what when. This involved assessment of what was needed and what was not. In some cases the caregiver had to search for people outside the family who could help with specific nursing activities.

Around half of the informants said they were next to the patient nearly all the time, frequently 24 hours per day. In most cases other family members or people close to the family in one or another way participated in caring. They assisted the caregiver with practical activities

or just being with the patient at times when the caregiver had to leave, but nevertheless the caregiver had to arrange or coordinate these supports. And sometimes the caretakers had to manage advice as well.

*My mom spent time with grandma, being together and speaking with her, but all nursing was on me. I am not afraid of that, I know what and how to do, even if Grandma was resisting. Mom said, you are bothering her too much, but I explained that I am doing what I have to do. We will have problems later if we don't do all cleaning and washing. (Granddaughter 48)*

*Thank God, we had a neighbor who was a nurse, retired. I asked her to come to do intrave-nous injections when Mom was sick with pneumonia. (Daughter 37)*

**4d. Being together: Providing emotional support.**   Caregivers' narratives revealed differ-ent levels of emotional relationship between caregiver and the terminally ill person. Emotional support was frequently noted as being particularly significant at the moment of diagnosis and the first days after. In most situations caregivers were the persons closest to the terminally ill patient. Both caregivers and patients expected to spend as much time as possible together. Caregivers told about such small simple things as playing musical instruments, reading, pray-ing together. Frequently they noted that they had spent time discussing themes that were important to the terminally ill person. An overload of responsibilities at this time of organizing care made this need for being together difficult to realize.

*Father got almost blind after chemotherapy. He couldn't do anything except listening to radio and TV. He wanted to communicate . . . . (Son 24)*

Two informants reported that they had had negative and painful relationship histories and that the patients' personalities were difficult. In these cases, they avoided emotional closeness with the patient, focusing on practical things they had to do.

*I took care of him as a zombie. He made me suffer so much during our life together. . . . But I did all things needed. (Wife 68)*

Existing health or emotional problems of other family members, especially those related to grieving, put additional responsibility on caregivers.

**4e. You have to stay there: Negative experience of hospitalization.**   All terminally ill patients cared for by participants in this study were hospitalized for shorter or longer periods, often more than one time. Caregivers reported that, owing to previous negative experiences of hospitalization, they spent large amounts of time staying at the hospital with the patient. In the informants' opinion, the content (what) and form (how) of institutional care didn't meet the needs of the terminally ill patients whom they cared for. Staying in the hospital involved some direct actions on the part of the caregiver (e.g., feeding and combing) and ensuring better (more pleasant) treatment from staff, even protecting the patient. One caregiver summed up this strong finding, stating that "If you are there, they [staff] treat the patient differently [bet-ter]. All know that."

*Well, I saw how they [nurses] treated patients . . . those who don't have somebody visiting him in the ward. How to say . . . very formally. For example, they bring food, put it on a cabinet next to bed and leave. In some time they come again, say, "You didn't eat anything" and take*

*it out. But the patient in that situation can't eat by himself, he needs somebody's help. (Son 48)*

*I understood it wasn't the first time she used those rude words screaming at her. Mom started to apologize to me for getting wet in bed. . . . But these rude words were a shock for Mom. I got lost, started trembling . . . I didn't know what to do. I called to my brother, I wanted to ask him what we should do. But we have never left her alone at hospital since that day. (Daughter 50)*

A common way of trying to protect patients and ensure better care was to give money to the staff at all levels. While the provision of such bribes "goes without saying," in that it is known to all and is tolerated, it was a particularly unpleasant part of protecting the patient. Along with bribes, informal networks of relationships with relatives, professionals, and influential persons could sometimes lead to access to services. Several participants wanted reassurance that their comments would not lead to difficulties for them or for the persons being paid or influenced. The line between a gift to an underpaid hospital employee and a bribe to receive service was ill-defined in some cases.

**4f. Exhausted physically and emotionally.**    The majority of informants reported that as the loved one's condition deteriorated, they experienced more and more physical and emotional exhaustion.

*My Mom was afraid to stay alone at home. So it became more and more difficult psychologically. Especially in nighttime. She asked me to come and be with her. I couldn't sleep normally . . . I felt I was getting crazy . . . 24 hours at home with her . . .. (Daughter 37)*

*I got pain in my back since I didn't know how to care her appropriately. (Wife 46)*

**4g. Confronting consequences.**    All informants openly talked about the consequences that such a burdensome role had on their well-being, as well as on their relationship with the person being cared for and her or his quality of life.

The multiplicity of nursing functions and many other role dimensions became a burden, a distinct hardship. The informants said that this role amounted to full, complete devotion to the person cared for, typically a loved one. Such devotion, as the stories of the informants show, is a renunciation of oneself, one's needs, interests, desires and even one's free time. One informant expressed the well-being of the main caregiver succinctly: "here is where the abyss is, really."

Most informants said that due to fatigue arising from physical, emotional, psychological exertion, they faced and survived certain periods of crisis. The informants overcame crises differently. Some cried or cried out somewhere else or in solitude, others took sleeping pills, and others struggled with the urge to escape somewhere:

*Something still had to be given up. For example, well, it is mostly related to your life and your free time, the satisfaction of some of your desires. . . . At first, I sometimes wanted to run, and I ran away, I left her. When I came back, she told me how scared she was when she was left alone, and she couldn't take care of herself. Then I tried not to run away. When it was hard to stay there, I prayed . . .. (Daughter 45)*

However, perhaps the most significant consequence of the multifunctionality of the main caregiver and the resulting physical, emotional and psychological fatigue could be found in the periods of deteriorating relationships with the cared loved one.

*Well, I didn't really want to betray him, although sometimes it seemed that I could possibly hurt him, yeah . . .. (Wife 71)*

*It was difficult for me and maybe I feel some kind of reproach that I said, oh how tired I am, because I was just very tired, and I didn't hold back . . . Probably he didn't feel dignified at moments like that . . . I don't know, maybe. (Wife 61)*

## Discussion

This study identifies four themes that describe the social role of main end-of-life caregiver in its social context. The themes also address the process of taking on the role of caregiver. Study participants shared their experiences and understandings from their own frame of reference, which was accepted as the context of the thematic analysis. One strength of the study is that taps the internal sense of identity in the role along with the surface persona that descriptions of caregivers are often limited to in research. The themes are not claimed to be universal, and experiences vary, but there was a consistency in the sample suggesting that the themes represent significant phenomena in the study's Lithuanian setting, and probably beyond. The findings come from accounts of volunteers, not from a random sample of the population of caregivers. Since any generalization from the findings of studies of this type have to be made on conceptual rather than statistical bases, the findings are particularly valuable as the building blocks for future examination of constructs and hypotheses related to the overall concern for quality of care and dignity at the end of life.

Informants shared that they became informal caregivers in part because of the poor quality, including the inaccessibility, of public medical facilities and services. These findings from the first theme (Inaccessibility and mistrust of public care services) are similar to the ones in previous research studies conducted in Lithuania which revealed that the mechanisms for providing necessary services were not yet sufficiently developed. Furthermore, there were no favorable conditions for independent living in the community, because care-at-home was not accessible [19, 33]. There have been local improvements in services that may not be consistent with some of the participants' experiences, but the circumstances that the participants lived in are likely to continue. Informal caregiving was seen as a medical necessity, in addition to any relational motives for caring for another. This theme involved a perception that was founded on experience. The remains of the authoritarian devaluation of the individual by professionals and government systems were evident to the caregivers, although for many this was a given that they were accustomed to, and there were exceptions. Professionals' and systems' failures to serve those in need of care meant that the main role of care had to be taken by someone else, informally, without training in preparation for the challenges involved. Informants did not share an expectation that all care should essentially be a state responsibility. Some informants shared their understanding of the difficulties faced by professional and non-professional medical staff. But in the face of difficult medical situations and diagnoses, they were frustrated that they could not get adequate necessary technical support, expert guidance, cooperation, or communication. Isolation rather than partnership was predominant.

A second theme (Moral obligations and responsibilities of immediate family and friends) states that one becomes a caregiver because it is the right thing to do morally, in certain circumstances. This theme was determined by family norms and other social norms and expectations as well as by the individual's own moral character. The position of the individual in a constellation of interpersonal and structural family relations might dictate whose moral responsibility it was to become the caregiver. There was not a ritual or formal calculus of who

would be the main caretaker, but there was typically a tacit understanding of how roles should develop to meet care needs. Filial and marital relationships were predominant determinants of this role assignment. The significance of the theme is that morality of caregiving was important to the caregivers, but it is not often included in research on caregiving. A comprehensive and sufficient caregiving partnership would acknowledge the role of professional support in matters of balancing formal and moral determinants of caregiving. According to previous systematic review studies, "the recognition of the value of the caregiver role may contribute to a positive caregiving experience and decrease rates of patient hospitalization and institutionalization" [34] which consequently might lead to more respectful and dignified communication between health professionals, patients and their caregivers and "supports the ethics of care and dignity conceptual frameworks" [35].

The third theme ("It's our generation": cultural aspect) emphasizes caregiver as a social role, consistent with earlier studies. The informants clearly articulated the power of culture as a determinant of role prescriptions and expectations. The theme shows that cultural traditions are not only material, they also are passed down in relational practices. One of the predominant traditions was that people from younger generations take care of older relatives. But this theme includes another important factor, the changing society and social context, especially the increasingly complex roles of the middle-aged women who do much of the caregiving. The theme of cultural traditions of what is expected of "our generation" is coupled with the acceptance of the caregiving role, even when it is added to the roles of wife, mother, worker, and community member, leading to more expectations than one can readily fulfill. In order to be complete, research on caregiving should account for the many roles that are filled simultaneously, in detail. Informants, for example, sometimes spoke of conversations with the people they cared for concerning whether or not to stop formal employment in order to be caregiver. This common conversation may become a focal point for working out fundamental relationship responsibilities at the end of life. The theme of generational responsibility for care was sometimes strained by contemporary role demands in other areas of life. Contemporary social and economic realities were sometimes acutely in conflict with cultural patterns. The issues of caregivers' confusions on sharing responsibilities with professional care providers, as well as their uncertainty regarding their role in care of severely ill patients, followed by burdens and loss of caregivers' dignity were also reported by recent Norwegian, Czech and Iranian qualitative studies [25, 36, 37].

The fourth theme (The caregiver feels responsible for everything) is linked to the mix of internal and external factors determining the caregiver role and its meaning. It builds on the substance of first three themes present. There was a clear sense of feeling "responsible for everything" that can be understood partly by listing the many facets of the role: practical nursing and learning how to do it, consultation with doctors and being the mediator of communication between doctor and patient, practical management of the logistics of care, providing emotional support, and exhaustion. These are described in a number of studies. But another element of the theme of being responsible for everything relates to the risk of what Nemati et al. referred to as disintegration, "a state of helplessness and turmoil in the caregiver's personal, emotional and family system" [6]. As the disease progressed and the demands of care became greater, the isolation increased as other social contacts withdrew. One set of acute examples of this phenomenon sometimes occurred in the various inpatient units where patients with incurable conditions were assigned. The perceived necessity of staying in the hospital around the clock with the patient to provide safety and to ensure even minimal care represented a major challenge to the caregivers.

The challenge was compounded by the need to provide informal payments, simply called bribes, for administrators and providers of each element of service that they performed. This

practice was a consistent reminder that being a patient or a caregiver was not valued in its own right, that payment was required even to go through the motions, that the value of the human being was reduced. Participants surfaced a Lithuanian version of what Garcia [38] referred to as the "open secret" of corruption as they experienced Lithuanian EoL care and services. The matter is more than corruption in the form of illegal advantages obtained in return for something of value. The practice of giving gifts, providing envelopes of cash, and showing gratitude in health care was very complex. The caregivers descriptions of the situation confirmed Praspaliauskiene's analysis [39]. The caregivers' difficulties around payment for care services is also consistent with the findings of Transparency International [40, 41]. While it has been argued that, given the massive prevalence in society of symbolic informal payments for medical care, society does not consider this to be a reprehensible and dangerous phenomenon, which should be punished by criminal law, but rather it is recognized as a social norm [42]. Norm or not, informal payments contributed to the sense of being responsible for practically everything to do with caregiving.

The level of importance of this theme transcends the constructs of caregiver burden and being overwhelmed. Being responsible for everything can take a toll on caregivers, rendering them into a state of feeling powerless, less able to understand themselves in their role as the agents of effective, compassionate care. The scarce institutional support for informal (family) caregivers and a lack of cooperation are relatively common for many other countries in the CEE region [23]. The successful development of end of life care services in Poland, while limited to cancer patients, is rather an exception from the rule that gaps in equal access to palliative care and integration of services. This represents the impact of political (social) changes to the caregiver role in the CEE countries including Lithuania [43]. In Lithuania psychosocial services were not easily accessible for the relatives of the patient. An earlier study showed that patients' family members and relatives, who were constantly caring for their patient, were particularly in need of services to allow rest from daily patient care [44]. Caregivers' health problems related to their role in patient care were also observed [45].

The results of the study confirm earlier findings that the role of family caregiver is not static with fixed obligations and clear outcomes [22, 46]. The correlates or determinants of the role are not all fixed. The process of becoming a caregiver involves taking on and evolving a set of role dimensions related to the situation and the individuals in it. Since there is a process, or series of sequential stages, involving a number of identified variables that determine the experience of caregiving, it is possible build a model to reduce the difficult aspects of the role and to improve the experience for the caregiver and the one cared for alike. These can be done at the micro level, as professional and non-professional teams do in some places in developing care plans. In fact, each individual case is its own entity, with its own cluster of role elements and meanings that are gleaned from enacting them. This would be the realm of sound clinical practice with a partnership orientation.

A broader way of thinking of the problems of caregivers and the consequences of the problems can be considered when caregivers are thought of as a population at risk in a given community. This public health orientation is based on demographics of aging populations in many societies and the number of people assuming the caregiving role. Not all caregivers have trying experiences, but enough do to merit their being considered as a population at risk. Public health modeling puts into play a range of research and intervention possibilities. Any organized attempt to influence the public's health behavior involves epidemiology of the conditions involved, specification of what would constitute desirable outcomes, clarification and education about the problems in terms that the public can understand, influencing the public's valuation of alternatives, and provision of the resources to allow change to happen. Policy and political commitment are necessary for the development of concrete resources and the public

health infrastructure to organize campaigns. Health services research is invaluable in this process, particularly the evaluation of different ways of framing policy, service delivery systems, specific professional practices, administrative discretion, and financing arrangements in terms of outcomes in well-being and dignity of those who are near death and those who care for them.

Caregivers have ideas based on their experiences of caring. Their ideas about what constitutes dignity would form a critical outcome that should be further investigated and operationalized. The idea of evidence-based practice derived from public health systems research should include the valid perspectives of all the stakeholders in order for them to be both agents of change and beneficiaries.

## Conclusion

The themes developed in one small study giving voice to the caregivers do not describe the full experience of caregiving at the end of life. However, they do represent important elements of the caregiving experience as it is experienced by caregivers. Recognition of caregiving experiences are essential in planning services and ways to deliver them that are meaningful and effective. The process of assuming the role of caregiver and evolving its meaning is unique in each instance, but there are commonalities that can be understood and that should be taken into account in practice, both clinical and public health. Many of categories and themes in this Lithuanian study are based on aversive experiences with current care systems. These may highlight elements of experience that are not well appreciated in research and service planning. Future research can further illuminate these, and particularly the significance of aversive and positive experiences of the process of becoming a caregiver in order to enhance the dignity of caregiving and thus of dying.

## Supporting information

**S1 File. Interview guide.**
(PDF)

**S2 File. Sample of anonymized interviews [in Lithuanian].** https://doi.org/10.5281/zenodo.6334994.
(DOCX)

**S3 File. Progression from sample of interview text to codes, categories, and theme for Theme 1.**
(DOCX)

**S4 File.**
(PDF)

**S5 File.**
(DOCX)

## Author Contributions

**Conceptualization:** Jolanta Kuznecovienė, Rūta Butkevičienė, W. David Harrison, Eimantas Peičius, Gvidas Urbonas, Kristina Astromskė.

**Data curation:** Jolanta Kuznecovienė, Rūta Butkevičienė, W. David Harrison, Eimantas Peičius, Gvidas Urbonas, Kristina Astromskė.

**Investigation:** Jolanta Kuznecovienė, Rūta Butkevičienė, W. David Harrison, Eimantas Peičius, Gvidas Urbonas, Kristina Astromskė.

**Methodology:** Jolanta Kuznecovienė, Rūta Butkevičienė, W. David Harrison, Eimantas Peičius, Gvidas Urbonas, Kristina Astromskė.

**Project administration:** Jolanta Kuznecovienė.

**Resources:** Rūta Butkevičienė, Eimantas Peičius, Gvidas Urbonas, Kristina Astromskė.

**Validation:** Jolanta Kuznecovienė, Rūta Butkevičienė, W. David Harrison, Eimantas Peičius, Gvidas Urbonas, Kristina Astromskė.

**Writing – original draft:** Jolanta Kuznecovienė, Rūta Butkevičienė, W. David Harrison, Eimantas Peičius, Gvidas Urbonas, Kristina Astromskė.

**Writing – review & editing:** Jolanta Kuznecovienė, Rūta Butkevičienė, W. David Harrison, Eimantas Peičius, Gvidas Urbonas, Kristina Astromskė.

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
