## [Decision Letter · Decision Letter 0]

16 Feb 2021

PONE-D-21-00726

What does it mean to be the main caregiver to a terminally ill family member in Lithuania: the caregivers’ perspectives

PLOS ONE

Dear Dr. Kuznecoviene,

Thank you for submitting your manuscript to PLOS ONE. After careful consideration, we feel that it has merit but does not fully meet PLOS ONE’s publication criteria as it currently stands. Therefore, we invite you to submit a revised version of the manuscript that addresses the points raised during the review process.

Reviewers have found that the topic of the research is interesting to the field. However there are some major issues that should be addressed. Please follow the COREQ guidelines for the presentation of qualitative research and include all the methodological information (required by reviewer 2) in the method section of the manuscript. 

We look forward to receiving your revised manuscript.

Kind regards,

Manuel Fernández-Alcántara, Ph.D.

Academic Editor

PLOS ONE

Journal Requirements:

Reviewers' comments:

Reviewer's Responses to Questions

**Comments to the Author**

1. Is the manuscript technically sound, and do the data support the conclusions?

Reviewer #1: Yes

Reviewer #2: Partly

2. Has the statistical analysis been performed appropriately and rigorously? 

Reviewer #1: N/A

Reviewer #2: Yes

3. Have the authors made all data underlying the findings in their manuscript fully available?

Reviewer #1: Yes

Reviewer #2: Yes

4. Is the manuscript presented in an intelligible fashion and written in standard English?

Reviewer #1: Yes

Reviewer #2: Yes

5. Review Comments to the Author

Reviewer #1: Strengths of the submission include the following:

1) Given the nascent nature of palliative care services in Lithuania, and the critical role that family caregivers play in providing care to those with life-limiting illnesses, it is important to examine and describe family caregiver experiences within this specific geographical context.

2) Overall, the paper is well organized and written.

3) The data exemplars provided help to illustrate and appear emblematic of the themes presented.

4) There is mention of a theoretical grounding for the study, i.e. role theory.

5) The sampling approaches and data analysis procedures are well described and there is some mention of the issue of rigor.

Areas requiring attention to strengthen the submission:

1) The authors do reference some family caregiver literature in their paper, but the manuscript lacks a critical synthesis of the large body of family caregiver research that has been done to date. I would have expected to see included the seminal works of Dr. Linda Kristjanson, for example. The authors do situate their findings vis a vis some extant work, but their findings are not that surprising, and it begs the question as to why they conducted the particular study that they did. I appreciate that their might be cultural variations, but the exact gap that the project purported to fill is not compelling.

2) To their credit, the authors identify role theory, particularly the thinking of Merton, as informing their work. The paper would have been stronger had they provided more information about the theory and its underlying assumptions, as well as a critique of its limitations. For example, not all family members will agree to undertake the caregiver role, and such a stance would be seen as deviant behavior. How does role theory account for this? To what extent does the use of role theory as a theoretical lens reinforce commonly held notions about how people should behave? This is particularly relevant given that the majority of caregivers in the study were women of a certain generation.

3) The biggest limitation in the paper is that the authors invoke the notion of dignity, and dignified care, but never define the construct, or integrate the well developed body of literature about dignity and end of life care in the paper. This is a significant gap.

4) The issue of corruption in paying to help ensure better care provision in institutions and/or to “move patients up in the queue is an important, albeit disturbing finding. The manuscript would have been strengthened had this finding been situated within literature that exists talking about the magnitude of the problem of corruption, and the responsibilities of researchers in all countries in helping to address it. (e.g. see Lancet 2019 https://www.thelancet.com/journals/lancet/article/PIIS0140-6736(19)32527-9/fulltext)

5) I appreciate the need to move to virtual interviews for data collection amidst the strain of the pandemic! Despite advantages including convenience and interactivity of collecting data virtually, the literature does document a number of ethical, practical, and interactional issues associated with the use of digital technologies. The paper would have been stronger had these been mentioned in the paper, and the authors’ experiences compared with them.

Reviewer #2: The topic itself on “What does it mean to be the main caregiver to a terminally ill family member in Lithuania: the caregivers’ perspectives” has the potential to become an issue of caregivers. The study conceptualized  caregivers’ understanding of the functions that they assume while being the main caregivers, and to understand how they experienced the consequences they confronted for filling the research gap. The paper deals with an important topic that is likely to be of interest to the researchers. In its current form, I may suggest the author modified the errors of details and enhance the final manuscript.

Introduction

(1)It is better to state more clearly about the objectives and research questions.

Materials and Methods

(1)Consider the guide for the presentation of qualitative studies (COREQ)

(2)It is better to state clearly why 33 caregivers included in the current study?

(3)The authors adopted qualitative methods in the current study. It is better to state clearly the logic of qualitative methods. It is better to provide a table about descriptive conditions about the interviewees.

(4)The authors should state clearly about the setting of data collection, number of participants that refused to participate or dropped out, field notes, data saturation.

Results

（1）The authors should state clearly why the paper include 4 themes of the main caregiver of a terminally ill family member and the meaning of the caregiver role.

（2）The authors should state about number of data coders, consistency between data and findings, software used, and participant feedback.

Discussions

(1)The discussion part should connect with the results part, at present, the themes in the part of results have not been stated in the discussion part.

6. PLOS authors have the option to publish the peer review history of their article (what does this mean?). If published, this will include your full peer review and any attached files.

Reviewer #1: No

Reviewer #2: No

---

## [Author Response · Author response to Decision Letter 0]

22 Apr 2021

Reviewer 1. We have incorporated all of your suggestions into our revision. They were very helpful. Our responses to your comments are presented in the file "Response to reviewers". Thank you.

Reviewer 2. We have incorporated all of your suggestions into our revision. Thank you for your help. They were very helpful. Our responses to your comments are presented in the file "Response to reviewers". Thank you.

---

## [Decision Letter · Decision Letter 1]

13 May 2021

PONE-D-21-00726R1

What does it mean to be the main caregiver to a terminally ill family member in Lithuania: the caregivers’ perspectives

PLOS ONE

Dear Dr. Kuznecoviene,

Thank you for submitting your manuscript to PLOS ONE. After careful consideration, we feel that it has merit but does not fully meet PLOS ONE’s publication criteria as it currently stands. Therefore, we invite you to submit a revised version of the manuscript that addresses the points raised during the review process.

Authors have addressed the suggestions and commentaries of reviewers. However, I have noted two minor points that should be addressed before final acceptance:

- Please modify the title of the manuscript to include the qualitative design. I will recommend something as " What does it mean to be the main caregiver to a terminally ill family member in Lithuania: a qualitative study".

- The first paragraph of the introduction (where authors state their objectives) should be placed at the end of the introduction (just before Material and Methods section)

We look forward to receiving your revised manuscript.

Kind regards,

Manuel Fernández-Alcántara, Ph.D.

Academic Editor

PLOS ONE

Journal Requirements:

Reviewers' comments:

Reviewer's Responses to Questions

**Comments to the Author**

1. If the authors have adequately addressed your comments raised in a previous round of review and you feel that this manuscript is now acceptable for publication, you may indicate that here to bypass the “Comments to the Author” section, enter your conflict of interest statement in the “Confidential to Editor” section, and submit your "Accept" recommendation.

Reviewer #2: All comments have been addressed

2. Is the manuscript technically sound, and do the data support the conclusions?

Reviewer #2: Yes

3. Has the statistical analysis been performed appropriately and rigorously? 

Reviewer #2: Yes

4. Have the authors made all data underlying the findings in their manuscript fully available?

Reviewer #2: Yes

5. Is the manuscript presented in an intelligible fashion and written in standard English?

Reviewer #2: Yes

6. Review Comments to the Author

Reviewer #2: I think the revised version is acceptable. The authors have made efforts on paper writing, especially the introduction part.

7. PLOS authors have the option to publish the peer review history of their article (what does this mean?). If published, this will include your full peer review and any attached files.

Reviewer #2: **Yes: **Yong TANG

---

## [Author Response · Author response to Decision Letter 1]

22 Jul 2021

Re: PONE -D-21-00726RI

10 June 2021

Manuel Fernández-Alcántara, Ph.D.

Academic Editor

PLOS ONE

Dear Dr. Fernández-Alcántara:

Thank you for your two minor points in the manuscript PONE -D-21-00726RI, previously entitled “What does it mean to be the main caregiver to a terminally ill family member in Lithuania: the caregivers’ perspectives.” 

We have addressed these two minor changes as requested and consider the manuscript complete to your standards. We are attaching a file with the new, revised and complete manuscript (filename: Manuscript), and a file with the manuscript clearly showing the tracking of the new changes (filename: Revised Manuscript with Track Changes).

Please note that these files show that the following changes have been made according to your guidance:

Requested change 1. “Please modify the title of the manuscript to include the qualitative design. I will recommend something as " What does it mean to be the main caregiver to a terminally ill family member in Lithuania: a qualitative study".”

Completed change 1. We have changed the title accordingly to “What does it mean to be the main caregiver to a terminally ill family member in Lithuania?: A qualitative study”.

Requested change 2. “The first paragraph of the introduction (where authors state their objectives) should be placed at the end of the introduction (just before Material and Methods section)”

Completed change 2. We have moved the specified text to the specified place in the manuscript, just before the Material and Methods section.

We appreciate your careful reviews and guidance in preparing the manuscript for publication and look forward to final notification of its acceptance.

Sincerely,

Jolanta Kuznecovienė, PhD

---

## [Editor Report · Decision Letter 2]

26 Oct 2021

PONE-D-21-00726R2What does it mean to be the main caregiver to a terminally ill family member in Lithuania: A qualitative studyPLOS ONE

Dear Dr. Kuznecoviene,

Thank you for submitting your manuscript to PLOS ONE. After careful consideration, we feel that it has merit but does not fully meet PLOS ONE’s publication criteria as it currently stands. Therefore, we invite you to submit a revised version of the manuscript that addresses the points raised during the review process.Please ensure that your decision is justified on PLOS ONE’s publication criteria and not, for example, on novelty or perceived impact.

We look forward to receiving your revised manuscript.

Kind regards,

Andrew Soundy

Academic Editor

PLOS ONE

Journal Requirements:

Additional Editor Comments (if provided):

• Please make sure the reader understands the philosphical underpinnings of descriptive thematic analysis – I have had a look at your reference 27 that states: “Applied thematic analysis as we define it comprises a bit of everything—grounded theory, positivism, interpretivism, and phenomenology—synthesized into one methodological framework” – personally I find that statement extremely odd because they mention, paradigms of positivism and interpretivism which are opposites alongside methodologies of grounded theory (but don’t name a type) and phenomenology (again no type is given). You will have to reference this because a reader needs to understand how you view the world using this approach and readers will expect a statement that considers a paradigm and methodology. To me it sounds like a pragmatic approach which uses different tools from paradigms and methodologies.

• Have a separate sub-heading for the interview guide and its development

• Have a separate sub-heading for procedures

• Reference 27 doesn’t mention ethnographic interviewing (line 170) it mentions grounded theory and phenomenology (but not types) – talking about ethnography may confuse the reader so consider rewording

• Have a section on sample size and how you decided when to stop interviewing

• Results seem to appear from around line 196 – 218 – please move to the start of the results

• Please include an audit trail with examples of analysis in a supplementary file readers need to have short example stages that they should go through should they wish to use the approach again - given reference 27's loose consideration to this it is important examples are provided

• Please include the original and any modified interview script in a supplementary file

• Please include a section on how quality was ensured – make sure it has its own sub-heading
---

## [Author Response · Author response to Decision Letter 2]

23 Nov 2021

Dear Dr. Soundy,

Regarding: POLS ONE Decision: Revision required [PONE-D-21-00726R2] – [EMID:0994672364td1bfe]

We appreciate your comments and recommendations for revision of our study, “What does it mean to be the main caregiver to a terminally ill family member in Lithuania?: A qualitative study.” We have made changes based on the revised manuscript that was sent in response to the first reviewers’ comments. The current changes are detailed with Track Changes, and a clean version of the revised manuscript is also included. 

Here is an account of what we have done in response to each of your points.

Editor’s point 1.

“• Please make sure the reader understands the philosphical underpinnings of descriptive thematic analysis – I have had a look at your reference 27 that states: “Applied thematic analysis as we define it comprises a bit of everything—grounded theory, positivism, interpretivism, and phenomenology—synthesized into one methodological framework” – personally I find that statement extremely odd because they mention, paradigms of positivism and interpretivism which are opposites alongside methodologies of grounded theory (but don’t name a type) and phenomenology (again no type is given). You will have to reference this because a reader needs to understand how you view the world using this approach and readers will expect a statement that considers a paradigm and methodology. To me it sounds like a pragmatic approach which uses different tools from paradigms and methodologies.”

We have deleted the first sentence of the first paragraph in the Materials and Methods section, and added the following, including a new reference that will be added to the References list [Pietkiewicz, I., and Smith, J.A. Czasopismo Psychologiczne – Psychological Journal, 20, 1, 2014, 7-14].:

Research Strategy

The logic and procedures of the study are shown in Table 1. The research team used the strategy of applied thematic analysis as presented by Guest, et al. [27] to structure the study. Applied thematic analysis is flexible and does not propose a methodological recipe. The strategy is integrative in that it allows for the pursuit of phenomenological understanding through the use of data collection and analysis methods derived in part from a positivistic tradition. Interview text is data observed and recorded in the real world, meaning that it is positivistic data. The analysis process was designed to seek commonalities of experience reflected in this data, and to apply an interpretive, phenomenological orientation to the systematic assessment and understanding of the essential meaning conveyed by the data. The team used the interpretive phenomenological analysis method advocated by Pietkiewicz and Smith. Although Pietkiewicz and Smith did not identify the use of positivistic data for interpretive elements explicitly, they describe the use of interview (and other empirical, positivistic) data as foundational elements to construct a phenomenological presentation of how people understand a situation. 

Editor’s point 2.

“• Have a separate sub-heading for the interview guide and its development”

The following new section has been added after the section entitled “Research Strategy”:

Procedures

Interview Guide

Qualitative methods of this type are particularly useful in capturing content and context that may not have been known or hypothesized a priori [27]. To find themes the team of six public health faculty members designed an interview guide that provided preliminary structure for data collection. The guide built upon the content of previous studies of caregivers’ behavior, many of which were framed explicitly or tacitly by the concept of social role. For example, Stajduhar et al. reviewed dozens of studies that emphasize the knowledge and skills of the caregiving role and how caregivers acquire them [11]. The intent of the guide was to explore the experiences of caregivers in fulfilling the caregiver role, including role elements concerning everyday life of caregiving, the needs of the terminally ill person and the family, patterns of communication and decision-making with professionals, and caregivers’ perceptions of a dignified end-of-life experience. Team members were sensitive to the changes that one might experience in the role over time. Interviewers used probes to ensure that the data were characterized by clarity, sensitivity to difficult topics, openness to important new topics, and explanations of inconsistencies [28]. 

Discussion to refine the early drafts of the guide and informal pre-tests led to the decision to reduce the number of stimulus questions and to frame them to allow for a broader range of responses. After analysis of the first interviews, the researchers supplemented the guide with emerging questions on topics about knowledge and use of social services and other formal and informal social supports, the aversion to and fear of using any sort of inpatient palliative care facilities, and the necessity to provide unofficial payments to medical personnel in order to receive any services. Consistent with the design of the study, these topics reflect specific Lithuanian realities that may be clearer in the country’s specific national context than they would be in different settings where theyare less apparent to practitioners, but still present. The questions stimulating by far the most valuable data were “How do you personally understand dignity at the end-of-life?” and “What factors ensure dignity at the end of life?”

Editor’s point 3.

“• Have a separate sub-heading for procedures”

This sub-heading appears now before Interview Guide

Editor’s point 4.

“• Reference 27 doesn’t mention ethnographic interviewing (line 170) it mentions grounded theory and phenomenology (but not types) – talking about ethnography may confuse the reader so consider rewording”

The term “ethnographic” has been deleted for clarity.

Editor’s point 5.

“• Have a section on sample size and how you decided when to stop interviewing”

A sentence has been added at the end of the first paragraph of the section entitled “Sample” and a new section entitled “Sample Size and Characteristics” has been added:

The sample allowed the researchers to emphasize the fundamental commonality of informant experience while allowing for variations in order to fully develop themes from the data.

Sample Size and Characteristics

The researchers concluded that substantial saturation of content categories was reached after 33 interviews, at which point interviewing stopped. This was determined by the research team’s high degree of agreement that data that were being consistently coded in specific groups or categories, and that new coding was not repeated significantly. The researchers concluded that substantial saturation of content categories was reached after 33 interviews.

Editor’s point 6.

“• Results seem to appear from around line 196 – 218 – please move to the start of the results”

We could not identify this point by line number. 

Editor’s point 7.

“• Please include an audit trail with examples of analysis in a supplementary file readers need to have short example stages that they should go through should they wish to use the approach again - given reference 27's loose consideration to this it is important examples are provided”

A new sentence and a reference to a Supplementary File with a sample of the process of have been added at the end of the “Data Analysis” section:

The process is illustrated with the development of Theme 1 in Supplementary file 2, Progression from sample of interview text to codes, categories, and theme.

A new section entitled “Quality Assurance” with a revised first paragraph have been added after the section entitled “Data Analysis”:

Quality Assurance

Four members of the research team discussed the data and its analysis at the beginning of this stage (Phase 3, Stage 2 from Table 1). Different insights and disagreements in understanding and interpretation were reconciled and the results were agreed upon. In order to strengthen trustworthiness of the findings, the other two team members independently reviewed the data and evolved themes at stage three. While the names of some codes, categories, and themes varied, there was a high degree of agreement on substance. The consistency and compatibility of findings between the data analysts made reconciliation possible with a high degree of concordance. The themes were also presented for discussion by medical professionals during a round-table seminar organized by the research team. Despite a number of digressions based on specific findings, the round-table group concurred that the findings were consistent with the realities that they faced or had experienced, even though they were understood from quite different viewpoints. Table 1. And Supplementary file 2 constitute an audit trail for the data analysis process.

Editor’s point 8.

“• Please include the original and any modified interview script in a supplementary file”

Supplementary File 3 presents a sample from the original transcripts. Full transcripts are available upon request.

Editor’s point 9. 

“• Please include a section on how quality was ensured – make sure it has its own sub-heading”

Covered in conjunction with Editor’s point 7.

We wish to leave Supplementary File 1 (Interview Guide) as it was filed with the original manuscript. We have revised Supplementary File 2, changing the title from “Coded Interview Data” to “Progression from sample of interview text to codes, categories, and theme.” And we have added a Supplementary File 3., entitled “Sample of original interview narrative text.”

We appreciate the opportunity to work with you on the publication and look forward to your response. It is important to note that our funding for publication of the article expires in December 2021, so it would be appreciated if we get your feedback quickly.

Sincerely, 

Jolanta Kuznecovienė

---

## [Editor Report · Decision Letter 3]

28 Feb 2022

What does it mean to be the main caregiver to a terminally ill family member in Lithuania: A qualitative study

PONE-D-21-00726R3

Dear Dr. Kuznecoviene,

We’re pleased to inform you that your manuscript has been judged scientifically suitable for publication and will be formally accepted for publication once it meets all outstanding technical requirements.

Kind regards,

Andrew Soundy

Academic Editor

PLOS ONE
---

## [Editor Report · Acceptance letter]

4 May 2022

PONE-D-21-00726R3 

What does it mean to be the main caregiver to a terminally ill family member in Lithuania?: A qualitative study 

Dear Dr. Kuznecoviene:

I'm pleased to inform you that your manuscript has been deemed suitable for publication in PLOS ONE. Congratulations! Your manuscript is now with our production department. 

Kind regards, 

on behalf of

Dr. Andrew Soundy 

Academic Editor

PLOS ONE